# Aspects of Quality of Life: Single vs. Mated People

**DOI:** 10.3390/bs14100954

**Published:** 2024-10-16

**Authors:** Menelaos Apostolou, Burcu Tekeş, Antonios Kagialis, Timo Juhani Lajunen

**Affiliations:** 1Department of Social Sciences, University of Nicosia, 46 Makedonitissas Ave, Nicosia 1700, Cyprus; 2Department of Psychology, Başkent University, Bağlıca Kampüsü, 06790 Ankara, Turkey; burcutekes@gmail.com; 3Department of Psychiatry, School of Medicine, University of Crete, 70013 Heraklion, Greece; med2p1080166@med.uoc.gr; 4Department of Psychology, Norwegian University of Science and Technology, 7034 Trondheim, Norway; timo.lajunen@ntnu.no

**Keywords:** singlehood, quality of life, life satisfaction, intimate relationships

## Abstract

Not having an intimate partner constitutes a common state in contemporary post-industrial societies. The current research aimed to address the question of whether single people score higher than mated people in various dimensions of quality of life. For this purpose, we employed quantitative research methods, measuring different aspects of quality of life that we treated as the dependent variables, with relationship status as the independent variable. In a sample of 1929 participants from Greece and Turkey, we found that relationship status was not significantly associated with physical health, relationships with friends and family, self-development, independence, recreation, or participation in social and communal activities. On the other hand, it was significantly associated with material goods, disposable income, social support, sexual satisfaction, and having children, with mated people scoring higher than single people. Despite using different methodologies for data collection, similar results were obtained in the two cultural contexts.

## 1. Introduction

Most cultures value long-term intimate relationships and the formation of a family [1,2]. However, in contemporary post-industrial societies, singlehood is on the rise, with several individuals not having an intimate partner [3,4]. Furthermore, many singles indicate that they are voluntarily in this status [5]. These observations raise the question of whether singles fare better or worse than people in intimate relationships. The current research aims to address this question by examining how the two groups differ in their satisfaction with various aspects of quality of life.

### 1.1. Quality of Life and Relationship Status

#### 1.1.1. Positive Emotions and Relationship Status

It could be reasonably argued that a higher quality of life is characterized by experiencing more positive emotions and fewer negative emotions (for further discussion on the link between wellbeing and intimate relationships, see [6,7]). For instance, it would be unlikely for people to be depressed but characterize their life as good. Taking an evolutionary perspective on human emotions, it has been argued that mated individuals, who are more likely to procreate, would enjoy higher emotional well-being than single people [8]. Consistent with this argument, several studies have found that married people are happier than unmarried people [9,10]. Nevertheless, these studies did not differentiate between participants who did not have an intimate partner and participants who were in an intimate relationship but not married, classifying both as single.

More recent studies have attempted to address this limitation by distinguishing between singles and individuals in a relationship but not married. In particular, one study employed a sample of 735 Greek-speaking participants and found that those who were in a relationship or married experienced fewer negative emotions, more positive emotions, and used a Polish sample and found that single individuals reported lower emotional well-being than partnered individuals, but they did not find any difference between the two groups in regard to mental health problems [11]. In addition, in a Polish university-student sample, one study found that single relationship status was related to greater romantic and family loneliness and to less perceived social support from significant others and family [12].

Furthermore, a different study found that not having an intimate partner was a significant risk factor for high levels of depression in men [13]. Other studies compared men who identified as incels (i.e., males who base their sense of identity around a perceived inability to form intimate relationships) with non-incels and found that the former experienced higher levels of depression, anxiety, and loneliness, as well as lower levels of life satisfaction [14,15,16]. Although more research is needed, the existing literature makes a strong case that, on average, people in an intimate relationship experience more positive emotions and fewer negative emotions than people who are single. Yet quality of life is a broader concept that includes not only emotional wellbeing but also external factors such as economic, social, and environmental conditions [17,18].

#### 1.1.2. Quality of Life Dimensions and Relationship Status

It has been argued that single individuals have more freedom, resources, and time to develop themselves. For example, they can pursue higher education, attend seminars, and gain experiences through travel [19]. Similarly, being single allows more time for physical exercise and sports, which can have a positive impact on health [20,21]. Additionally, single individuals can focus on developing their network of friends, potentially resulting in more and higher-quality friendships [22,23]. In turn, having close friends has been associated with higher life satisfaction [24]. Singles are also more likely to have better relationships and spend more time with their relatives [25]. We will examine these arguments in more detail.

To begin with, maintaining an intimate relationship requires a significant commitment of resources such as time (being there for your partner) and money (buying presents, for example). Furthermore, individuals in intimate relationships do not have the same freedom to do as they wish because they must consider the impact of their actions on their partners. Therefore, it is possible that they may be constrained in pursuing their goals and personal development. Consistent with this argument, one study asked a sample of Greek-speaking participants to indicate the advantages of being single and found that participants rated “More time for myself”, “Focus on my goals”, and “No one dictates my actions” as among the most important advantages [26]. Yet we could not locate any study that has directly examined whether single individuals have a better capacity to develop themselves compared to those in committed relationships.

Regarding physical activity and better health, one study examined panel data from a sample of 13,496 American men and women and found that unmarried individuals engaged in more physical exercise than married participants [20]. Nevertheless, a limitation of this study, as with others in the field, is that it did not differentiate between singles and individuals who are in an intimate relationship but not married, grouping them all as “unmarried”—referred to as the “differentiation problem”. Another study attempted to understand the priorities of singles in their lives using American and Korean samples and found that singles placed a high priority on being healthy [21]. Nevertheless, their sample did not include participants in committed relationships as a reference group. Therefore, we cannot exclude the possibility that individuals in an intimate relationship place an equal or even higher priority on their health compared to single individuals. In general, studies suggest that married people enjoy better health compared to individuals in other relationship status groups [10]. For instance, a meta-analysis including 34 studies with over two million participants found that married individuals were less likely to suffer from cardiovascular diseases [27]. Another meta-analysis including 15 studies and over 800,000 participants found that being married was associated with a reduced risk of dementia compared to being widowed or lifelong single [28]. 

Regarding friendships and relationships with relatives, one study examined the responses of 25,185 participants in the USA and found that unmarried individuals tended to have fewer friends than married people, although the effect of relationship status was relatively small [29]. A different study used panel data from the USA and found that unmarried individuals were more likely to frequently stay in touch with, provide help to, and receive help from parents, siblings, neighbors, and friends compared to married individuals [23]. However, both studies suffered from the differentiation problem. On the other hand, another study used a nationally representative dataset in the Netherlands that differentiated between dating and non-dating unmarried participants [30]. The study found that the number of friends and contacts with friends declined when people started dating or began living with their partners and increased when they went through a divorce. Additionally, some studies have found that unmarried individuals exchanged more support with their parents than married individuals [25,31], while others did not find a significant relationship between marital status and support or contact [32,33]. 

One study examined a dataset collected in the USA in the 1980s, which included a measure of satisfaction that people derived from specific aspects of their lives, including friends, relatives, home, neighborhood, job, financial situation, and leisure pursuits [34]. They found that married individuals reported the highest scores, followed by individuals in cohabiting relationships, steady dating relationships, casual dating relationships, and individuals who dated infrequently or not at all. Nevertheless, the study did not separately examine the different aspects associated with life satisfaction. Most importantly, it did not have a clearly defined single category—the most relevant category had 74 participants who dated infrequently or not at all in the last month.

### 1.2. The Current Study

Although several studies have examined the impact of relationship status on various positive life outcomes, they have been hindered by the differentiation problem. Consequently, our understanding of whether single people differ from those in relationships in different dimensions of quality of life remains incomplete. The current research aims to contribute to closing this gap in our knowledge by examining the association of relationship status with positive life outcomes in two different cultural settings: Greece and Turkey. The two cultures share a common history, as the geographical area of Greece was part of the Ottoman Empire for about four centuries. As a consequence, there is some cultural overlap between the two nations, but there are also substantial differences. For instance, the majority of the Greek population is Christian Orthodox, while the majority of the Turkish population is Muslim. For a more extensive discussion of the similarities and differences, see [35].

Previous research has focused on five dimensions related to quality of life, namely (a) physical and material well-being, (b) relations with other people, (c) participation in social, community, and civic activities, (d) personal development and fulfillment, and (e) recreation [36,37]. Therefore, our study compared single and mated individuals in these dimensions. Furthermore, as discussed earlier, it has been argued that freedom is a significant advantage of being single [19]. Previous research has also found that one reason people indicate for not being in an intimate relationship is the desire to have sex with different partners [38], suggesting that relationship status predicts sexual satisfaction, which appears to be the case [21,39]. Additionally, it has been argued that one advantage of being in a relationship is having someone to provide reliable support [40]. Similarly, for many individuals, having children is a key life objective [41], indicating that it impacts how they perceive the quality of their lives. As freedom, sexual satisfaction, support, and children are important aspects of life quality, our study aimed to examine the effect of relationship status on these dimensions as well. In addition, individuals in an intimate relationship are more likely to have children than those who are single. Therefore, it is important to control for parental status; otherwise, the relationship status variable could inadvertently serve as a proxy for whether participants have children.

Our study was designed to address the question whether relationship status predicts different aspects of quality of life. It is important to note that our research is exploratory, as we are interested in investigating how single and partnered individuals differ in several positive life outcomes without making any directional hypotheses. Moreover, our study predominantly employed subjective rather than objective measures for positive outcomes, as subjective measures better capture quality of life. For instance, some individuals may be satisfied with having three close friends, while others may prefer having six, indicating that the number of close friends may not be the best indicator of people’s satisfaction in the friendship aspect of quality of life.

## 2. Methods

### 2.1. Participants

The study was performed at a private university in the Republic of Cyprus and in a private university in Turkey and has received ethics clearance from the respective boards of each university. We employed a convenience sampling method, where participants were recruited by forwarding the link on social media (Facebook, Instagram, Twitter) and to students and colleagues with the request to also share it with their networks. The only requirement for participation was to be at least 18 years old. For the Greek sample, participants received no monetary or other compensation, while for the Turkish sample, some students received course credits for taking part. In total, our sample included 1929 participants. The Greek sample included 555 participants (301 women, 250 men, 2 who indicated ‘other’, and 2 who did not indicate their sex). The mean age of women was 33.4 (*SD* = 10.8), and the mean age of men was 34.0 (*SD* = 12.0). In addition, 42.8% of the participants were single, 31.0% in a relationship, 21.8% married, and 4.3% indicated their marital status as ‘other’. The mean age of singles was 29.1 (*SD* = 10.6), that of participants in a relationship was 26 (*SD* = 7.1), and that of married participants was 42.1 (*SD* = 10.2). Furthermore, 25.4% of the participants indicated that they had children. The data were collected in February and March 2023.

The Turkish sample included 1374 participants (920 women, 448 men, 5 who did not indicate their sex, and 5 who indicated ‘other’). The mean age of women was 33.1 (*SD* = 12.2), and the mean age of men was 33.3 (*SD* = 13.1). Furthermore, 42.8% of the participants were married, 28.1% single, 25.7% in a relationship, and 3.4% indicated their marital status as ‘other.’ The mean age of singles was 27.4 (*SD* = 10.2), that of participants in a relationship was 24.7 (*SD* = 6.6), and that of married participants was 41.6 (*SD* = 10.3). Additionally, 41.1% of the participants indicated that they had children. The data were collected in March 2023. All data are available here: https://osf.io/jz54m/?view_only=37fc9ae059a64099b4e02c3fc9745551, accessed on 21 May 2024.

### 2.2. Materials

The questionnaire was created in Google Forms and consisted of four sections, available in both Greek and Turkish. In the first section, we employed the Quality of Life Scale (QOLS) to assess satisfaction with six dimensions of quality of life [36,37,42]. The instrument included 16 items such as “Material comforts home, food, conveniences, financial security” and “Close friends” that participants rated on a seven-point Likert scale ranging from 1 (Terrible) to 7 (Delighted). The Cronbach’s alpha for this instrument was 0.89. The six dimensions and the 16 items can be found in Table 1.

In the second part, we measured social support using the Social Support Questionnaire (SSQ) [43]. The instrument consisted of 12 items such as “When I need suggestions on how to deal with a personal problem, I know someone I can turn to” that participants rated on a four-point Likert scale ranging from “Definitely True” to “Definitely False”. A higher total score indicated higher social support. The original QOLS and SSQ instruments were in English and were translated into Greek and Turkish using the back translation method. The Cronbach’s alpha for this instrument was 0.84.

The third part of the questionnaire asked participants to indicate their satisfaction with their sexual life, housing, and means of transportation (e.g., I have money left to save) (Table 2). Participants provided their answers on a seven-point scale ranging from 1 (not at all satisfied) to 7 (very satisfied). They were also asked to assess whether they had enough money to cover their needs and save, using a seven-point scale ranging from 1 (strongly disagree) to 7 (strongly agree). Participants further assessed their health using a one-item instrument on a scale from ‘Poor’ to ‘Excellent’. In particular, participants were asked “In general, would you say your health is:” and their answers were recorded in the following scale: “5—Excellent”, “4—Very good”, “3—Good”, “2—So and so”, “1—Bad”. Additionally, participants indicated the number of close friends they had, using a scale of none, 1–2, 3–5, or more than 5. These two questions were adopted from the WHOQOL-BREF instrument (https://www.who.int/tools/whoqol/whoqol-bref accessed on 23 January 2023).

In the fourth part, demographic information was collected, including biological sex (woman, man, other), age, relationship status (single—not in an intimate relationship, in a relationship, married, other), and whether they had children (yes, no). The order of presentation of the different sections was randomized across participants.

### 2.3. Statistical Analysis

For the purposes of our analysis, we performed a series of ANCOVA tests. In particular, the dimension of interest was entered as the dependent variable and the relationship status as the independent variable. Participants’ sex, sample (Greek, Turkish), and having children (Yes, No) were entered as categorical and participant’s age as continuous covariates, respectively. We also performed logistic regression, with the number of close friends entered as the dependent variable, and participants’ relationship status as the independent variable. The rest of the variables entered as covariates as above. This analysis was repeated with the having children entering as the dependent variable. Note that, at least in the cultures under investigation, marriage constitutes a formal bond between the parties involved, which entails entitlements and obligations mandated by law. Conversely, being in an intimate relationship is an informal arrangement without legal consequences, which can be more easily dissolved. Given the differences between these groups, in our analysis, we distinguished between participants who were married and those who were in a relationship rather than collapsing both into one category. The analysis was performed using SPSS version 28 software package.

## 3. Results

We performed 25 tests in total, so in order to avoid the problem of alpha inflation, we can apply Bonferroni correction setting the alpha level to 0.002 (0.05/25). Accordingly, the reader may not consider significant any results with a higher *p* value. Beginning with the aspects of quality of life presented in Table 1, with respect to “Physical and material well-being”, there was a significant main effect of relationship status, with married participants reporting higher satisfaction than single participants. Yet, as indicated by the effect size, the difference was small. From Table 2, we can see that, when asked specifically about their satisfaction with their housing, participants who were married or in a relationship indicated higher satisfaction than single participants. Similarly, when asked about their satisfaction with their means of transportation, married indicated a higher satisfaction than single participants. In both cases, the effect size of relationship status was small. With respect to satisfaction with health, the relationship status had no effect (Table 1). This was also the case when participants were asked specifically to assess their health status (Table 2). In terms of satisfaction with family relationships, married participants gave higher scores than single participants, but the effect was not significant if Bonferroni correction was applied. With respect to the “Relations with other people” dimension, for having and rearing children, married participants gave significantly higher scores than participants who were single. Similarly, with respect to close relationships with a significant other, married and participants in a relationship gave significantly higher scores than participants who were single, with the difference being substantial. On the other hand, relationship status was not significantly associated with satisfaction with close friends. In the same vein, logistic regression analysis found no significant effect of relationship status on how many close friends one has (*p* = 0.403).

Moving on to the “Participation in social, community, and civic activities” dimension, relationship status was not significantly associated with satisfaction with helping others and participating in organizations. For the “Personal development and fulfillment” dimension, relationship status was not significantly associated with learning, understanding oneself, or expressing oneself creatively. With respect to job satisfaction, participants who were married or in a relationship gave significantly higher scores than participants who were single, with the effect being small. With respect to recreation, relationship status was not significantly associated with socializing, with reading or observing entertainment, or with participating in active recreation. Similarly, relationship status was not significantly associated with the “Independence” dimension.

From Table 2, we can see that participants who were in an intimate relationship or married were significantly more likely to be satisfied with their sexual life than participants who were single, with the effect of relationship status being substantial. With respect to having enough money to cover one’s needs, married participants gave higher scores than single participants. Similarly, with respect to having enough money to save, participants who were married or in a relationship gave higher scores than single participants. Additionally, with respect to social support, participants who were in a relationship or married gave significantly higher scores than participants who were single, with the effect size of the relationship status being small. Moreover, the results of logistic regression indicated that there was a significant main effect of relationship status on having children [*χ^2^*(2, *N* = 1791) = 386.45, *p* < 0.001]. More specifically, married participants were 26.8 times more likely than single participants to have than not to have children. Yet there was no significant difference between participants in a relationship and single. Finally, in the tests above, we did not detect any interactions between relationship status and sex, indicating that the effect of the former on this aspect of quality of life was similar across sexes.

### Sample and Children Effects

From Table 1 and Table 2, we can see that, in most cases, there was no significant main effect of sample, and in cases where it was significant, it was small. For the number of friends, the sample had a significant effect [*χ*^2^(3, *N* = 1791) = 103.19, *p* < 0.001]. In particular, Greek-speaking participants were 5.05 time more likely than Turkish participants to have 1–2 close friends and 3.08 times more likely to have 3–4 close friends than more than 5 close friends. Furthermore, we did not detect any interactions between relationship status and sample, which indicates that our findings were similar in the Greek and the Turkish samples. In addition, there was no significant main effect of sample on children (*p* = 0.016). Moreover, children did not predict any of the dimensions associated with quality of life. There was one exception, namely over the ‘Having and rearing children’ facet, where participants who had children indicated a higher satisfaction (*M* = 6.0, *SD* = 1.3) than participants who did not (*M* = 3.3, *SD* = 2.0).

## 4. Discussion

In the current study, we investigated the association of relationship status with several dimensions of quality of life. We found that, for physical health, relationships with friends and family, self-development, independence, recreation, and participation in social and communal activities, relationship status had no effect. On the other hand, mated individuals scored higher than single individuals in terms of material goods, disposable income, social support, sexual satisfaction, and having children.

Starting with physical and material well-being, participants in relationships indicated higher satisfaction than single participants. One possible explanation is that individuals in relationships share expenses, resulting in higher disposable income that can be allocated to increasing material comfort, such as buying a better car. Still, the effect sizes were small, suggesting that material well-being is primarily influenced by other factors, such as personality and education, rather than relationship status. Additionally, individuals in relationships did not report higher satisfaction with their physical well-being or health status compared to single individuals. Previous research has found an association between marriage and better health outcomes, but these studies often suffered from the differentiation problem. Moreover, they focused on objective health outcomes, while our study measured subjective satisfaction and evaluation of health. It is also possible that the positive effects of being in an intimate relationship on health may appear later in life and were not detected in our sample, which had a limited number of older participants. For instance, single people tend to experience more negative emotions than mated individuals such as anxiety [8], which can lead to increased consumption of unhealthy foods [44] or smoking [45], which have negative health effects in the long run [46]. 

Regarding relations with other people, there was a trend for participants to indicate higher satisfaction with their interactions with relatives. Yet relationship status was not a significant predictor of satisfaction with close friends or the number of close friends. Previous research has found that the number of friends decreases when individuals enter into a relationship and increases when they exit a relationship [30]. In our study, we focused on close friends rather than friends in general, which may explain why we did not detect such an effect. Our findings suggest that relationship status has little influence on the number of close friends individuals have. In terms of satisfaction with close relationships with a significant other, single individuals reported considerably lower scores than those in relationships. This finding is consistent with research findings that indicate that many singles are not satisfied with their relationship status and would prefer to have an intimate partner [5]. 

Regarding participation in social, community, and civic activities, relationship status did not differentiate between single and partnered individuals in terms of satisfaction. Furthermore, relationship status had no effect on the other facets of personal development and fulfillment dimension, except for work satisfaction, where individuals in relationships reported higher satisfaction compared to single individuals, although the effect size was small. One possible explanation is that having a secure partner allows individuals to devote more energy to advancing their careers, which could explain why participants in relationships enjoyed higher job satisfaction. For example, studies find that people spend considerable resources such as time and money in enhancing their appeal as mates [47] and on trying to locate prospective mates [48] which, once they have secured intimate partners, could become available to achieve other goals.

Relationship status had no effect on the recreation and independence dimensions of quality of life. This finding was surprising, as one might expect single individuals, who do not have to consider their partners in decision-making, to be more satisfied with their independence than individuals in relationships. The effect size was zero, suggesting that this result is unlikely to be spurious. One possible interpretation is that individuals in intimate relationships, due to having more resources and receiving support from their partners, have more freedom to pursue their interests and life’s goals, which may balance the perceived loss of freedom associated with being in a relationship. For instance, one study suggested that married individuals could better advance on their careers partially due to the economic support offered by their partners [49]. Further research is needed to understand the role of relationship status in this dimension. Our findings also suggest that individuals in relationships are better off in terms of having someone to provide support. Yet the effect size was small, indicating that an intimate partner is not the sole or primary source of support, with friends and relatives also being important sources.

Moving on, we found that individuals in relationships reported much higher sexual satisfaction than single individuals, a finding which is also consistent with the findings of other studies [21,39]. Previous studies have found that one reason for being single is the desire to have sex with different partners [38], suggesting that singlehood could be associated with higher sexual satisfaction. People vary considerably in their desire for sexual variety [50], but most tend to prefer monogamous relationships [51]. Therefore, it is possible that a minority of singles who desire sexual variety and are able to attract different partners experience high sexual satisfaction. The remaining singles who do not fall into this category, either preferring a long-term sexual partner or desiring multiple sexual partners but unable to attract them, may experience lower sexual satisfaction.

Raising children requires considerable resources, such as money and time, which people must inevitably divert from other pursuits. Consequently, we would expect that parents need to make compromises, potentially leading to a deterioration in certain aspects of their quality of life. For instance, they might have less time for volunteering, self-development, socializing, and entertainment. Therefore, it initially seems puzzling that, apart from the ‘Having and rearing children’ aspect, there was no significant main effect of children on any other aspects of quality of life we investigated. One explanation is that, in the current study, we measured subjective satisfaction with different life aspects. When people have children, they anticipate making compromises in some areas of their lives, so these compromises have little impact on their satisfaction. For example, people expect to have less leisure time when they have children, so they do not become dissatisfied with having less time for recreation. Future research should investigate whether these anticipated compromises explain why parenthood status does not predict satisfaction with the aspects of quality of life examined here.

For many individuals, having children is a key objective in life, and they are willing to go to great lengths to achieve it (e.g., IVF, surrogate mothers, etc.). Accordingly, children are an important aspect of quality of life [37,42]. Having an intimate partner is not a prerequisite for having children, as casual sexual contacts could suffice. Additionally, technological advancements, such as IVF and surrogate mothers, can allow people to have children without sexual contact. Moreover, while raising children requires substantial resources, social protection systems can alleviate some of the burdens for single parents. Nevertheless, after controlling for age, married individuals were 26 times more likely than single individuals to have children, indicating that being married plays a significant role in whether someone has children or not. There are several reasons for this effect. First, although technological advancements allow people to have children without an intimate partner, these methods can be expensive and out of reach for most individuals. Similarly, the cost of raising children is substantial, and in most cases, government support is insufficient to cover these expenses. Therefore, people often need to be in a long-term committed relationship or marriage in order to have children. It should be noted that a reverse causation effect may also be at play here, as individuals who do not want to have children may be less likely to get married compared to those who desire children.

In revisiting the debate on whether single life is better than being in a relationship, one argument in favor of single life is that individuals have more freedom and possibly more resources to pursue what is important to them, including personal development, maintaining their relationships with friends, and strengthening their network of friendships [3,19]. Nevertheless, our results do not support these claims, at least within the Greek and Turkish cultural contexts. The most likely explanation for our findings is that these dimensions are influenced by factors other than relationship status. In different words, whether someone is in a relationship or not does not seem to significantly constrain or enable individuals to achieve positive outcomes such as personal development or having close friends. One factor that could potentially play a role is the quality of the relationship. For instance, people who are in a good relationship may enjoy higher emotional wellbeing [6,52,53], which in turn can enable them to work more effectively in other areas of their life, such as forging close friendships. On the other hand, poor relationship quality could decrease people’s emotional wellbeing, leading them to perform poorly in other areas of life. Thus, future research needs to extend our work by examining the impact of relationship quality on different aspects of quality of life.

Our findings can have practical value for people who wonder about the impact that a change in their relationship status would have on their lives. For individuals who fear that entering an intimate relationship would negatively affect their satisfaction with aspects of life such as recreation, self-development, independence, and social interactions, our study suggests that this is unlikely to be the case. Conversely, for people who enter into an intimate relationship anticipating that it will improve their quality of life in these dimensions, our study indicates that this is also unlikely to happen. However, entering an intimate relationship is likely to enable people to enjoy a better financial situation, more sexual satisfaction, and increased support. It is important for readers to be aware that we refer to statistical averages, meaning that what happens on average may not occur in individual cases. For instance, although people in relationships are on average better off financially than single people, it is possible that an intimate relationship could lead to a deterioration of one’s financial situation (e.g., having a partner who absorbs considerable monetary resources without providing any in return).

Our study has several limitations. To begin with, we employed self-report instruments, which are subject to several biases, including participants providing inaccurate answers. Additionally, we employed non-probability samples, which may limit the generalizability of our findings to the broader population (but see [54]). Moreover, our study is correlational, so causality can only be indirectly inferred. This limitation could be addressed by future longitudinal studies, which could examine how various aspects of quality of life change as people enter and exit intimate relationships. Furthermore, we had access to two different countries, namely Greece and Turkey, and we sampled from both in order to investigate whether the effect of relationship status on different aspects of quality of life was similar across different cultural settings. We indeed found similar results between the two cultural contexts; however, this similarity could be partially due to their proximity and overlapping history. Future research should include more culturally diverse samples to examine the cultural consistency and variation of these findings. In addition, relationship status may interact with other variables, such as personality traits or desire for sexual variation, in predicting satisfaction with different facets of quality of life. These variables were not measured in the present study and should be the focus of future research. Similarly, our study focused on subjective satisfaction with aspects of life such as income, employment, and education but did not actually measure these variables, an endeavor that future studies need to undertake. Moreover, as discussed above, when examining the association between relationship status and quality of life outcomes, it is important to consider not only whether someone is in an intimate relationship or not but also the quality of the relationship. In the present study, this dimension was not measured, and future studies need to account for this limitation.

There has been a considerable amount of work comparing single and partnered individuals across various positive life outcomes. However, much of this research has suffered from the differentiation problem, while studies that did not focused almost exclusively on emotional differences between the two groups. Yet quality of life is multidimensional, so there is a gap in the literature regarding whether single and partnered individuals differ in various aspects of quality of life, which the current study has attempted to address. This endeavor becomes even more important in light of the argument put forward by some scholars that, aside from emotions, singles perform better than partnered people in many of these dimensions. Our findings indicate that, for several aspects of quality of life, neither singles nor partnered individuals have a distinct advantage, while for several others, single individuals fare worse than those in relationships. Further work is needed to better understand the relationship between quality of life and relationship status.

## Figures and Tables

**Table 1 behavsci-14-00954-t001:** Relationship status, sample, and children effects on different aspects of quality of life.

Aspects of Quality of Life	Relationship Status					Sample			
	Single	in a Relationship	Married			Greece	Turkey		
Physical and Material Well-Being	M (SD)	M (SD)	M (SD)	*p*-Value	η_p_^2^	M (SD)	M (SD)	*p*-Value	η_p_^2^
Material comforts home, food, conveniences, financial security	4.80 (1.66) m	4.94 (1.62)	5.09 (1.56) s	0.002	0.007	5.12 (1.46)	4.86 (1.47)	0.009	0.004
Health—being physically fit and vigorous	4.60 (1.63)	4.67 (1.60)	4.79 (1.54)	0.662	0.000	4.61 (1.55)	4.72 (1.161)	0.222	0.001
**Relations with other people**									
Relationships with parents, siblings and other relatives—communicating, visiting, helping	4.81 (1.67) m	5.01 (1.64)	5.25 (1.56) s	0.037	0.004	4.99 (1.63)	5.06 (1.64)	0.113	0.001
Having and rearing children	3.10 (2.04) m	3.77 (1.91)	5.76 (1.66) s	<0.001	0.013	3.96 (2.22)	4.48 (2.217)	0.801	0.000
Close relationships with spouse or significant other	2.99 (1.96) m,r	5.72 (1.50)s	5.36 (1.68) s	<0.001	0.106	4.11 (2.20)	4.84 (2.05)	0.167	0.001
Close friends	5.28 (1.67)	5.41 (1.59)	5.50 (1.5)	0.83	0.003	4.98 (1.72)	5.54 (1.52)	0.002	0.005
**Participation in social, community, and civic activities**									
Helping and encouraging others, volunteering, giving advice	5.26 (1.55)	5.19 (1.56)	5.53 (1.49)	0.884	0.000	4.66 (1.63)	5.63 (1.4)	<0.001	0.017
Participating in organizations and public affairs	3.62 (1.68)	3.67 (1.67)	4.10 (1.80)	0.617	0.001	3.76 (1.71)	3.83 (1.77)	0.135	0.001
**Personal development and fulfillment**									
Learning—attending school, improving understanding, getting additional knowledge	4.89 (1.62)	4.82 (1.70)	4.94 (1.66)	0.597	0.001	4.90 (1.62)	4.86 (1.68)	0.305	0.001
Understanding yourself—knowing your assets and limitations—knowing what life is about	5.03 (1.48)	5.13 (1.48)	5.47 (1.35)	0.137	0.002	5.25 (1.39)	5.22 (1.47)	0.434	0.000
Work—job or in home	4.23 (1.75) m,r	4.49 (1.77) s	4.98 (1.64) s	<0.001	0.008	4.59 (1.80)	4.56 (1.74)	0.014	0.003
Expressing yourself creatively	4.77 (1.60)	4.89 (1.55)	5.17 (1.44)	0.922	0.000	4.65 (1.57)	5.08 (1.51)	0.008	0.004
**Recreation**									
Socializing—meeting other people, doing things, parties, etc.	4.50 (1.80)	4.80 (1.65)	4.75 (1.70)	0.247	0.002	4.48 (1.67)	4.76 (1.74)	0.073	0.002
Reading, listening to music, or observing entertainment	5.19 (1.53)	5.28 (1.48)	4.99 (1.57)	0.091	0.003	5.09 (1.51)	5.15 (1.54)	0.121	0.001
Participating in active recreation	4.11 (1.88)	4.19 (1.92)	4.17 (1.88)	0.446	0.001	4.02 (1.85)	4.18 (1.92)	0.144	0.001
**Independence**									
Independence, doing for yourself	5.04 (1.69)	4.97 (1.75)	4.90 (1.68)	0.960	0.000	4.97 (1.69)	4.97 (1.70)	0.776	0.000

Note. For the relationship status variable, “m” indicates a significant difference with the “married” category, “r” with the in a relationship category, and “s” with the single category.

**Table 2 behavsci-14-00954-t002:** Relationship status and sample effects on additional aspects of quality of life.

Additional Aspects Associated with Quality of Life	Relationship Status					Sample			
	Single	in a Relationship	Married			Greece	Turkey		
	M (SD)	M (SD)	M (SD)	*p*-Value	η_p_^2^	M (SD)	M (SD)	*p*-Value	η_p_^2^
How satisfied are you with your sex life?	2.68 (1.87) m,r	5.03 (1.79) s	4.74 (1.88) s	<0.001	0.070	3.79 (2.09)	4.23 (2.13)	0.326	0.0001
How satisfied are you with your place of residence?	4.61 (1.75) m,r	4.84 (1.80) s	5.07 (1.68) s	0.002	0.007	4.82 (1.68)	4.83 (1.78)	0.259	0.001
How satisfied are you with your means of transportation (car, public transport, etc.)?	4.36 (1.96) m	4.54 (2.00)	5.20 (1.70) s	0.002	0.007	4.86 (1.89)	4.70 (1.92)	0.188	0.001
I have enough money to meet my needs.	4.54 (1.89) m	4.66 (1.74)	5.00 (1.74) s	0.003	0.007	4.45 (1.84)	4.86 (1.78)	0.157	0.001
I have money left to save	3.35 (2.06) m,r	3.54 (2.05) s	3.82 (2.05) s	0.009	0.005	3.50 (2.10)	3.61 (2.07)	0.461	0.000
In general, would you say your health is:	3.15 (0.94)	3.16 (0.97)	2.96 (0.91)	0.400	0.001	3.55 (0.93)	2.90 (0.89)	<0.001	0.028
Social support	3.49 (0.63) m,r	3.70 (0.58) s	3.73 (0.54) s	<0.001	0.015	3.49 (0.63)	3.69 (0.57)	<0.001	0.006

Note. For the relationship status variable, “m” indicates a significant difference with the “married” category, “r” with the in a relationship category, and “s” with the single category.

## Data Availability

All data are available here: https://osf.io/jz54m/?view_only=37fc9ae059a64099b4e02c3fc9745551 accessed on 21 May 2024.

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
