# Peer review of "Aspects of Quality of Life: Single vs. Mated People"

_behavsci, 2024, doi:10.3390/bs14100954_

Round 1

Reviewer 1 Report

Comments and Suggestions for Authors

Introduction

The literature review was exceptionally well-written and clearly organized. Relevant research was summarized; however, some more recent citations about romantic relationship status could improve the rationale. It was important to include the differentiation problem in prior research and the authors did so appropriately. The life satisfaction outcomes are clearly rationalized by connecting them to existing research.

I think the literature review is a good set up for a descriptive study that the author(s) performed. However, within family science research, the nature of relationships (e.g., quality, instability) is well established as having more predictive power for variables such as health and well-being. There is also a lack of theory supporting the rationale.

Methods

 The use of self-perceived satisfaction is an appropriate way to measure these outcomes, and the authors point this out in the current study section. However, it seems a contradiction that they said that a person could be satisfied with 3 friends rather than 6, yet this variable was measured using a number. Predicting that a person has few or many friends doesn’t really illuminate or illustrate their satisfaction or perceptions of friendships.

A major limitation (which I noticed is acknowledged in the discussion) is the lack of inclusion of relationship quality which is shown to be influential in the association between relationship status and personal outcomes, particularly health.

Results

In Table 1, I believe that a label would clarify the results about children and it seems that the means and SDs are missing for the children sample comparison. On a related note, the research questions and rationale do not specifically articulate that there would be an exploration of childfree people and people with children so I’m not sure why these results are presented in the table. If the aim is to understand 1) relationship status, 2) culture, and 3) child status on life satisfaction variables, then this makes sense. However, the way it was described, I was surprised to find these results. Either they can be removed, or the literature review should be revised to set up this portion of the study.

The results about children and sample are described on page 10. Consider removing the data about children from the table and only include in-text, especially if this is not the focus of the investigation. I would also recommend moving the final sentence in that paragraph before you describe the children results.

Discussion

The author(s) made clear connections to existing literature as they reviewed the findings of this study. Unanticipated findings were examined within the context of existing research and the implication of this work was well articulated. However, there were several interpretations that warrant review.

I think it’s an overstatement that because the single status participants “many singles are dissatisfied with their singlehood status and desire an intimate partner.” The questionnaire did not ask single participants if they desired a partner. If they did not have a significant relationship with a person, then of course they can’t be satified with a non-existent relationship. This is the same as people without children rating their satisfaction with rearing children low. It is likely a floor effect and should be interpreted as such. Similarly, the measures described didn’t ask single people about whether or not they were involuntarily single or not, so be cautios about overstating this finding.

Regarding the paragraph about sexual satisfaction, the final sentences about support seem to fit better in the previous paragraph.

I think if the results of the impact of having children on quality of life are included in as much detail as is now, there should be more interpretation in the discussion. I was left wondering, how is it possible that having children didn’t have any main effects on quality of life except the obvious, ‘having and rearing children’ aspect? How does this align with other research? The more obvious finding that married people are more likely to have children is explained thoroughly in the discussion but the other information about children and quality of life is missing.  

Comments on the Quality of English Language

English language is excellent and would benefit from only minor edits. 

Author Response

We would like to thank you very much for considering our work, and for providing us with valuable feedback that enabled us to improve our manuscript. Please see below how we have addressed all your concerns and recommendations.

Introduction

The literature review was exceptionally well-written and clearly organized. Relevant research was summarized; however, some more recent citations about romantic relationship status could improve the rationale. It was important to include the differentiation problem in prior research and the authors did so appropriately. The life satisfaction outcomes are clearly rationalized by connecting them to existing research.

-Thank you for your kind words. We aimed to include the most relevant studies in the area. However, please let us know if there is any specific paper you would like to reference and discuss.

I think the literature review is a good set up for a descriptive study that the author(s) performed. However, within family science research, the nature of relationships (e.g., quality, instability) is well established as having more predictive power for variables such as health and well-being. There is also a lack of theory supporting the rationale.

-Yes, we agree and in this revision we have discuss the issue of the quality of the relationships (please see below).

Moreover, you are right the present study is largely atheoretical. Let us explain why: As the rates of singlehood increase, the debate on whether being single or in an intimate relationship has become more popular. The proponents of singlehood make the argument that if people are single have more time and freedom to work on improving themselves, and build more meaningful relationships with friends and family. We think that these arguments are potentially misleading, as they are not backed by empirical evidence. This was our motivation for our study – we wanted to examine whether single individuals are indeed better off in these domains than mated individuals. Our findings clearly indicate that they are not, and we think that these findings need to be present in the literature and in an open-access journal as they would also be of interest to the wider audience. Thus, you can see that our study was not driven by a theoretical perspective, but we do not think this is a problem given the nature of the question that the present study aimed to answer.

Methods

 The use of self-perceived satisfaction is an appropriate way to measure these outcomes, and the authors point this out in the current study section. However, it seems a contradiction that they said that a person could be satisfied with 3 friends rather than 6, yet this variable was measured using a number. Predicting that a person has few or many friends doesn’t really illuminate or illustrate their satisfaction or perceptions of friendships.

-Good point! Actually, the focus of our study was subjective satisfaction with close friendships (see the main instrument of our study in Table 1). However, as there were explicit arguments in the literature that single people have more friends than mated people. Accordingly, we thought that it would be a good idea to measure both satisfaction with and the actual number of friend one has, and investigate whether relationship status predicts both dimensions.

A major limitation (which I noticed is acknowledged in the discussion) is the lack of inclusion of relationship quality which is shown to be influential in the association between relationship status and personal outcomes, particularly health.

-You make a good point, and we agree. Following your suggestion, this limitation is now reported in the text (Discussion, end of second paragraph from the end).

Results

In Table 1, I believe that a label would clarify the results about children and it seems that the means and SDs are missing for the children sample comparison. On a related note, the research questions and rationale do not specifically articulate that there would be an exploration of childfree people and people with children so I’m not sure why these results are presented in the table. If the aim is to understand 1) relationship status, 2) culture, and 3) child status on life satisfaction variables, then this makes sense. However, the way it was described, I was surprised to find these results. Either they can be removed, or the literature review should be revised to set up this portion of the study.

The results about children and sample are described on page 10. Consider removing the data about children from the table and only include in-text, especially if this is not the focus of the investigation. I would also recommend moving the final sentence in that paragraph before you describe the children results.

-Yes good point. Investigating the effect of children status on quality of life was not an objective of our study. Yet, we had to control for this variable for the following reason: People in an intimate relationship are more likely than people who are single to have children. Consequently, if we did not control for children, relationship status could act as a proxy of parental status, which would bias our results. Following your suggestion, we now report this rational in the introduction (The current study, end of second paragraph from the end). Also, you are right, as parental status is not the main objective of the study, we have removed the children effects from Tables 1 and 2 and we now discuss them only in the text (Results). Moreover, following your suggestion, we have moved the last sentence earlier in the text (Results, 3.1 Sample and children effects).

Discussion

The author(s) made clear connections to existing literature as they reviewed the findings of this study. Unanticipated findings were examined within the context of existing research and the implication of this work was well articulated. However, there were several interpretations that warrant review.

I think it’s an overstatement that because the single status participants “many singles are dissatisfied with their singlehood status and desire an intimate partner.” The questionnaire did not ask single participants if they desired a partner. If they did not have a significant relationship with a person, then of course they can’t be satified with a non-existent relationship. This is the same as people without children rating their satisfaction with rearing children low. It is likely a floor effect and should be interpreted as such. Similarly, the measures described didn’t ask single people about whether or not they were involuntarily single or not, so be cautios about overstating this finding.

-Yes, good point. Following your suggestion, we have now rephrased the sentence referring to relationship satisfaction (Discussion, third paragraph, last sentence). We have also deleted the sentence about satisfaction with children, as it could be confusing.

Regarding the paragraph about sexual satisfaction, the final sentences about support seem to fit better in the previous paragraph.

-Yes, we agree. Following your suggestion, we have moved these sentences to the previous paragraph (Discussion, fifth paragraph, last sentence).

I think if the results of the impact of having children on quality of life are included in as much detail as is now, there should be more interpretation in the discussion. I was left wondering, how is it possible that having children didn’t have any main effects on quality of life except the obvious, ‘having and rearing children’ aspect? How does this align with other research? The more obvious finding that married people are more likely to have children is explained thoroughly in the discussion but the other information about children and quality of life is missing.

-That is an excellent point! Given that children affect many parameters in life, we would expect to have an impact on the aspects of quality of life investigated here, and yet they do not! Actually, this is unlikely to be spurious effect, as it was consistent in both the Greek and the Turkish cultural context. In addition, the effect sizes were very small or zero, indicating that this was not an issue of statistical power (in any case, our sample was large). We think that one possible reason behind the lack of these effect is that people in the current study, we have measured subjective satisfaction with different aspects of life. When people have children, they expect that they will make compromise in some areas of their lives, so these compromises have little effect in their satisfaction. For example, people expect that when they have children, they would have less leisure time, so they do not become dissatisfied with having less time for recreation. Following your suggestion, this is now discussed in the text (Discussion, seventh paragraph).

We would like to thank you very much once more for your feedback, and we hope that you will find our revisions to your satisfaction.

Reviewer 2 Report

Comments and Suggestions for Authors

The article addresses an interesting and current topic. However, some parts should be improved.

In general:

- Probably a revision of the English is needed, not always understandable.

Abstract:

- The authors should specify the type of study. What design? What analysis? What model or type of analysis did they apply?

- The last sentence (these findings were generally consistent across the two cultural contexts) is very vague and not very true, since the authors allude to the cultural difference as a limitation.

Introduction:

The authors speak almost interchangeably of quality of life (a concept that is usually used mostly in case of chronic diseases) and well-being.

It would be better to differentiate these two aspects and integrate the literature on well-being, using additional sources.

For example:

- Van Dierendonck, D., & Lam, H. (2023). Interventions to enhance eudaemonic psychological well‐being: A meta‐analytic review with Ryff's Scales of Psychological Well‐being. Applied Psychology: Health and Well‐Being, 15(2), 594-610.;

- Esposito, C., Di Napoli, I., Agueli, B., Marino, L., Procentese, F., & Arcidiacono, C. (2021). Well-being and the COVID-19 pandemic: A community psychology systematic review. European Psychologist, 26(4).

- Paradisi, M., Matera, C., & Nerini, A. (2024). Feeling important, feeling well. The association between mattering and well-being: A meta-analysis study. Journal of Happiness Studies, 25(1), 4.

The Current Study:

In this paragraph, it should be clearly stated what the aim of the study was and whether the authors advanced specific hypotheses.

Methods

Participants:

- The authors say "For the Greek sample, participants received no monetary or other compensation, while for the Turkish sample some students received course credits for taking part.". Did the authors examine whether this difference had an effect on the results?

- The authors report many statistics of the participants in the text. These could be a bit limited (since they are already present in a Table).

- It should be specified that the nature of the sampling is not probabilistic, but convenience.

Measeres:

- It would be useful to report some examples of items for each of the instruments used.

- The reliability level for each instrument should be reported.

Statistical Analysis

- Specify which program was used for the statistical analyses.

Results

- The authors say "We performed 25 tests in total...". Given the large number of analyses to be performed, wouldn't it have been more useful to apply the Structural Equation Model? It would have perhaps also given greater validity to the results.

- The paragraph is a bit confusingly written, so it is not immediately clear what the main result of the study is.

Discussion

- Be more explicit about the limitations of the study.

- Discuss more about the possible practical implications.

Comments on the Quality of English Language

Moderate editing of English language required.

Author Response

The article addresses an interesting and current topic. However, some parts should be improved.

We would like to thank you very much for considering our work and for providing us with valuable feedback that enabled us to improve our manuscript. Please see below how we have addressed all your concerns and recommendations.

In general:

- Probably a revision of the English is needed, not always understandable.

We have revised the manuscript so as to improve the readability.

Abstract:

- The authors should specify the type of study. What design? What analysis? What model or type of analysis did they apply?

Yes, following you suggestion, we have added this information in the abstract.

- The last sentence (these findings were generally consistent across the two cultural contexts) is very vague and not very true, since the authors allude to the cultural difference as a limitation.

Following your suggestion, we have rephrased the sentence to make it more accurate.

Introduction:

The authors speak almost interchangeably of quality of life (a concept that is usually used mostly in case of chronic diseases) and well-being.

It would be better to differentiate these two aspects and integrate the literature on well-being, using additional sources.

That is a good point. Although the terms wellbeing and quality of life are used interchangeably, quality of life is a more broader term. This has to be made explicit, as in the current research we aimed to examine the aspects of quality of life other than emotional wellbeing. Following your suggestion, we now differentiate between the two terms (1.1.1 Positive emotions and relationship status, last paragraph, last sentence).

The Current Study:

In this paragraph, it should be clearly stated what the aim of the study was and whether the authors advanced specific hypotheses.

Following your suggestion, in the beginning of the last paragraph of the introduction we state specifically the purpose of the present study, and we also state that was not designed to test directional hypotheses.

Methods

Participants:

- The authors say "For the Greek sample, participants received no monetary or other compensation, while for the Turkish sample some students received course credits for taking part.". Did the authors examine whether this difference had an effect on the results?

Good point. Although the students received credit for taking part, for reasons of anonymity, this information was not recorded in the dataset, so such comparison cannot be made. We do not have any reasons to believe however that it has affected our results.

- The authors report many statistics of the participants in the text. These could be a bit limited (since they are already present in a Table).

In the current revision, we have made some changes in this section, namely we discuss the impact of having children, which is not discussed in the tables. In general, if this is fine with you, we would like to keep briefly referring to the results presented in the table in the text, to enable a better understanding for the readers who may not be familiar with the statistical analysis we have performed.

- It should be specified that the nature of the sampling is not probabilistic, but convenience.

Done! (2.1 Participants, first paragraph, second sentence)

Measures:

- It would be useful to report some examples of items for each of the instruments used.

Done! (2.2 Materials)

- The reliability level for each instrument should be reported.

Done! (2.2 Materials)

Statistical Analysis

- Specify which program was used for the statistical analyses.

Done! (2.3 Statistical analysis, last sentence)

Results

- The authors say "We performed 25 tests in total...". Given the large number of analyses to be performed, wouldn't it have been more useful to apply the Structural Equation Model? It would have perhaps also given greater validity to the results.

We do not think that a structural equation model would be appropriate here, as we aimed to test the direct effect of relationship status on specific aspects of quality of life.

- The paragraph is a bit confusingly written, so it is not immediately clear what the main result of the study is.

Following your suggestion, we revised the paragraph to present the results more clearly.

Discussion

- Be more explicit about the limitations of the study.

Following your suggestion, we now state specifically that our study has several limitations (Discussion, second paragraph from the end, first sentence). We have also expanded the relevant part to add additional limitations (Discussion, second paragraph from the end).

- Discuss more about the possible practical implications.

Yes, good idea. Following your suggestion, we now discuss some practical implications in the text (Discussion, third paragraph from the end).

We would like to thank you very much once more for your feedback, and we hope that you will find our revisions to your satisfaction.

Round 2

Reviewer 1 Report

Comments and Suggestions for Authors

Thank you for considering my feedback and for your thoughtful responses. I appreciate the revisions and clarifications you made throughout the manuscript. The adjustment related to parent status, clarification regarding the atheoretical nature of the study, and the description of the limitation of omitting relationship quality all strengthen the manuscript. I look forward to seeing the finalized version of your work.

Comments on the Quality of English Language

Only minor English language issues that can be remedied with editing.